

# Surgical *vs.* transcatheter aortic valve replacement in patients over 75 years with aortic stenosis: sociodemographic profile, clinical characteristics, quality of life and functionality

Víctor Fradejas-Sastre[1,2,3], Paula Parás-Bravo[1,4], Manuel Herrero-Montes[1,4], María Paz-Zulueta[1,5], Ester Boixadera-Planas[6], Luis Manuel Fernández-Cacho[1], Gabriela Veiga-Fernández[2,3], Maria Elena Arnáiz-García[7] and Jose María De-la-Torre-Hernández[2,3]

[1] Nursing Department, Universidad de Cantabria, Santander, Cantabria, Spain
[2] Interventional Cardiology and Hemodynamics Unit, Hospital Universitario Marqués de Valdecilla, Spain, Santander, Cantabria, Spain
[3] Research Group in Cardiovascular, Instituto de Investigación Sanitaria Valdecilla (IDIVAL), Santander, Cantabria, Spain
[4] Nursing Research Group, Instituto de Investigación Sanitaria Valdecilla (IDIVAL), Santander, Cantabria, Spain
[5] Research Group in Health Law and Bioethics, Instituto de Investigación Sanitaria Valdecilla (IDIVAL), Santander, Cantabria, Spain
[6] Servei d'Estadística Aplicada, Universidad Autonoma de Barcelona, Barcelona, Cataluña, Spain
[7] Cardiac Surgery Service, Hospital Universitario de Salamanca, Spain, Salamanca, Castilla y Leon, Spain

Corresponding author
Paula Parás-Bravo,
paula.paras@unican.es

## ABSTRACT

**Background:** Aortic valve stenosis (AVS) affects 25% of the population over 65 years. At present, there is no curative medical treatment for AVS and therefore the surgical approach, consisting of transcatheter aortic valve replacement (TAVR) or surgical aortic valve replacement (SAVR), is the treatment of choice.

**Methodology:** The aim of this study was to analyze the sociodemographic and clinical characteristics, quality of life and functionality of a sample of patients with AVS over 75 years of age, who underwent TAVR or SAVR, applying standard clinical practice. A prospective multicenter observational study was conducted in two hospitals of the Spanish National Health System. Data were collected at baseline, 1, 6 months and 1 year.

**Results:** In total, 227 participants were included, with a mean age of 80.6 [SD 4.1]. Statistically significant differences were found in terms of quality of life, which was higher at 1 year in patients who underwent SAVR. In terms of functionality, SAVR patients obtained a better score ($p < 0.01$). However, patients who underwent TAVR began with a worse baseline situation and managed to increase their quality of life and functionality after 1 year of follow-up.

**Conclusion:** The individualized choice of TAVR or SAVR in patients with AVS improves patients' quality of life and function. Moreover, the TAVR procedure in patients with a worse baseline situation and a high surgical risk achieved a similar increase in quality of life and functionality compared to patients undergoing SAVR with a better baseline situation.

# INTRODUCTION

Despite the rise in infectious diseases, including the recent pandemic caused by the severe acute respiratory syndrome coronavirus 2 (SARS-CoV-2), circulatory system diseases remain the leading cause of death in Spain, accounting for 23% of the total (and a rate of 112.2 deaths per 100,000 inhabitants) according to recent data from the *National Institute of Statistics (2020)*. The World Health Organization (WHO) estimates that 17.9 million people die each year as a result of cardiovascular diseases (*World Health Organization (WHO), 2021*). Population aging and lifestyle habits contribute to the epidemic of cardiovascular diseases, which fail to distinguish by sex, age, race, or geographical location. Throughout this article we will focus on valvular conditions, specifically aortic stenosis (AS). This condition can be valvular, supravalvular or subvalvular, although valvular aortic stenosis is the most common (AVS). Calcified or degenerative AS is the most frequent valve disease in the western world and requires the most intervention (*Ferreira-González et al., 2013*). It affects 25% of the population >65 years of age with 2–9% of this population experiencing some degree of AS (*Tokarek et al., 2015*, *2016*; *Dziewierz et al., 2018*; *Olszewska et al., 2017*). The prevalence of calcified for degenerative AS is expected to increase in the coming years with the population ageing.

The latest update by the valvular heart disease guidelines published in 2021 (*Vahanian et al., 2022*), based on recent studies, advocates early intervention in asymptomatic patients with severe AS, since it has been shown that early intervention in certain subgroups improves the survival of these patients at follow-up (*Kang et al., 2020*).

Classically, surgery was the only approach, however, in the last 20 years, the great technological advances and the development of new procedures of structural interventionism with the implantation of aortic prostheses *via* transcatheter, have represented a major revolution (*Tokarek, Dziewierz & Dudek, 2021*; *Vahanian et al., 2022*).

Generally speaking, TAVR is considered the best option in patients aged ≥75 years or those who are inoperable or at high surgical risk, as assessed by the classic risk scales (EUROSCORE II and STS (Society of Thoracic Surgeons) score >8%); provided that transfemoral implantation can be per-formed. In contrast, in patients aged <75 years or at low risk, with scores EUROSCORE II and STS scores <4% in which transfemoral access is not feasible and who are operable, surgery is the best option. In the remainder of cases, both options are valid (*Nashef et al., 2012*; *O'Brien et al., 2009*).

Although both interventions have been extensively studied, patients' perceptions of their quality of life and functionality have received much less attention.

Disease activity indices are not always good predictors of patients' quality of life; therefore, it is also necessary to evaluate the patient's perceived state of well-being (*Arostegui Madariaga & Núñez Antón, 2008*) and their functionality.

The classic pivotal studies of TAVR (PARTNER and COREVALVE) obtained clear benefits in terms of morbidity and mortality with improvement in symptoms and quality of life. Differences were also observed in terms of speed of improvement and functionality (*Adams, Popma & Reardon, 2014*; *Reynolds et al., 2012*). However, the TAVR patients included in these studies had high surgical risk and were older. They began with very low quality of life scores, and therefore the reported results were overestimated. At the current study, we focus on studying patients within the same age range who received TAVR *vs.* SAVR to remove affects arising from age.

In view of the above, the aim of this study was to examine the sociodemographic profile, clinical characteristics, quality of life and functionality in a sample of patients with AS over 75 years of age, who underwent TAVR or SAVR, applying routine clinical practice in two hospitals of the Spanish National Health System.

## MATERIALS AND METHODS

### Study design

A multicenter, interprovincial cohort study with two groups (TAVR and SAVR) was carried out. Two hospitals of the Spanish National Health System participated: the Hospital Universitario Marqués de Valdecilla (HUMV) (Cantabria, Spain) and the Hospital Universitario de Salamanca (HUSA) (Castille and León, Spain).

The two hospitals participating in the study used the same treatment protocol and were similar in terms of structure, staff and organizational characteristics. Nevertheless, prior to the study, a pilot test was carried out with the first 30 patients to detect and resolve any difficulties.

Patient inclusion in the study was performed prospectively and consecutively between January 2017 and August 2018 (Fig. 1).

### Study subjects

The study subjects were patients over 75 years of age diagnosed with severe aortic stenosis indicated for TAVR or SAVR intervention.

Inclusion criteria: Eligible patients were all those diagnosed with severe AS, over the age of 75 years, who underwent TAVR, with or without previous percutaneous coronary intervention, or SAVR, with or without aortocoronary bypass grafting.

Exclusion criteria: Patients in the acute phase of the disease, with severe cognitive impairment or any kind of sensory deficit that could hamper administration of the study instruments (*e.g.*, visual or auditory deficits), as well as those who did not speak Spanish.

### Variables

Data were collected using an *ad hoc* questionnaire with the following variables: (1) sociodemographic profile, (2) cardiovascular risk factors, (3) previous cardiac pathology, (4) previous noncardiac pathology, (5) baseline situation, (6) quality of life (SF-36, EuroQol-5D), and (7) functionality (Barthel Scale).

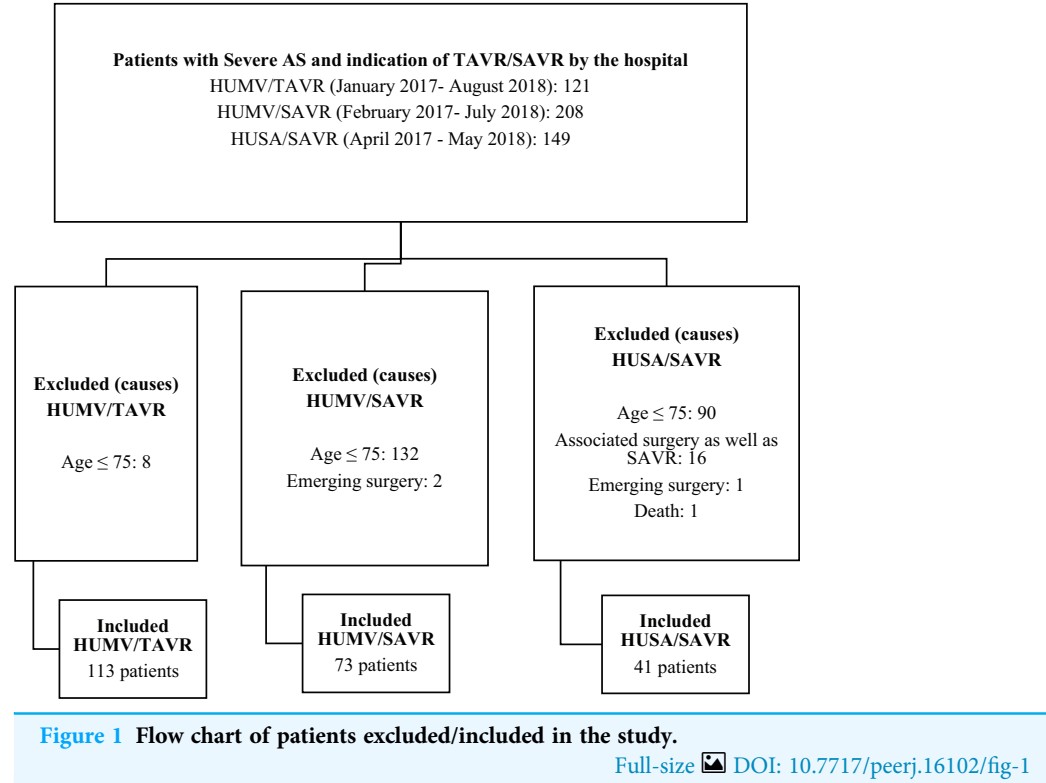

**Figure 1 Flow chart of patients excluded/included in the study.**

## Measurement tools
### Quality of life questionnaires

1. The 36-Item Short Form Health Survey questionnaire (SF-36)

The SF-36 was developed in the United States in the early 1990s for use in the Medical Outcomes Study (MOS) (*Ware & Sherbourne, 1992*). This is a generic scale that provides a profile of health status. It has proven useful for assessing health-related quality of life in the general population and in specific subgroups, comparing the burden of many different diseases, detecting the health benefits produced by a wide range of different treatments, and assessing the health status of individual patients. It is a generic instrument that contains 36 items that make up eight dimensions of the state of health and provide a profile of it. The eight dimensions evaluate: Physical Function, Physical Role, Bodily Pain, General Health, Vitality, Social Function, Emotional Role and Mental Health. The items and dimensions of the questionnaire provide scores that are directly proportional to the state of health; these scores are transformed into a scale that ranges from zero (the worst state of health for that dimension) to one hundred (the best state of health) (*Ware, 2000*).

2. EuroQol five-dimension scale (EuroQol-5D)

This is a generic Health-Related Quality of Life measurement instrument which, like the SF-36, can be used both in relatively healthy individuals (general population) and in groups of patients with different pathologies (*Badia et al., 1999*). The individual self-rates

their health status, first according to levels of severity by dimensions (descriptive system) and then using a visual analog scale for a more general evaluation. The descriptive system contains five dimensions of health (Mobility, Self Care, Usual Activities, Pain/discomfort, and Anxiety/depression) and each of them has three levels of severity (no problems, some problems or moderate problems, and severe problems). In this part of the questionnaire, the individual must mark the level of severity corresponding to their state of health in each of the dimensions, referring to the same day that the questionnaire is completed. In each dimension of the EuroQol-5D, the severity levels are scored between 1 and 3. Administration time is approximately 2–3 min.

### Functional assessment

1. Barthel Scale

The Barthel Scale assesses the patient's capacity for independence in self-care (*Mahoney & Barthel, 1965*). This is a generic measure that assesses the level of independence of the patient regarding the performance of certain basic activities of daily living, where different scores and weightings are assigned according to the ability of the subject examined to carry out these activities. It has ten parameters: feeding, bathing, dressing, personal hygiene, deposition, urination, toilet use, transfer from chair to bed, walking and steps. Each of them measures the person's capacity for independence or dependence on it. The total score for maximum independence is 100, and that for maximum dependence is 0. Changes are scored out of 5.

## Procedure

Patients diagnosed with severe AS were referred to the Interventional Cardiology Service for catheterization. After the procedure, the most appropriate treatment was discussed in a clinical session. Once the treatment was proposed and agreed with the patient, the person was invited to participate in the study. Those who agreed to participate were asked to provide written informed consent.

Follow-up was carried out for 1 year with four assessments:

– Pre-procedure assessment: patients were admitted 1 day before the procedure in the case of TAVR and 2 days before in SAVR. In both cases, on the same day of admission, an investigator went to the room to carry out the questionnaires. The interview was always conducted in person and preferably in the presence of a family member or caregiver who knew the patient well.

– Post-procedure assessment: an appointment was made with the patient and their family to carry out these three interviews (at 1 month, 6 months and 1 year) at their home or at the hospital if it coincided with the scheduling of another consultation.

## Statistical analysis of data

The data analysis included an initial descriptive analysis. For categorical and discrete variables, the contingency tables of each variable according to TAVR/SAVR are presented,

this result is complemented with the contrast, using Pearson's Chi-Squared Test for comparisons, or alternatively, Fisher's exact test when more than 20% of the cells presented a number of expected cases less than or equal to 5. For continuous variables, means were estimated with their standard deviation or medians and interquartile ranges (IQR) in the case of asymmetrical distributions. The comparison of quantitative variable between groups was done using Student t-test (in case of normal distribution of quantitative variable) or Mann-Whitney-Wilcoxon test (in case of non-normal distribution). The normality condition was previously verified using the Shapiro-Wilk test. The alpha error was set at 0.05 and all p's were bilateral. All statistical analyses were performed with the SAS v9.4, SAS Institute Inc., Cary, NC, USA. The raw measurements are available in the File S1.

### Ethical and legal considerations

This project was approved by the Ethics and Clinical Research Committee on July 11, 2014. (Protocol code: 22/2014). The participants were informed of the study, and provided their written consent to participate. The information was treated confidentially as regulated by Law 41/2002, of November 14, 2002, which regulates patient autonomy and the rights and obligations regarding clinical information and documentation (*Goverment of Spain, 2002*). In accordance with this law, this consent could be revoked at any time.

## RESULTS

Before the end of the 1-year follow-up, 15 patients in the TAVR group and 11 patients in the SAVR group died, representing a loss rate of 11.45%.

### Sociodemographic and clinical characteristics (Table 1)

The total sample consisted of 227 participants from the Hospital Universitario Marqués de Valdecilla (Santander, Cantabria, Spain) and Hospital Clínico Universitario de Salamanca (Salamanca, Castille and Leon, Spain) between January 2017 and August 2018. Of the 227 patients, 113 (50%) underwent TAVR and 114 (50%) underwent SAVR. Of the total patients, 82% were included from the Hospital Universitario Marqués de Valdecilla, whereas 18% belonged to the Hospital Universitario de Salamanca, in the latter case, all 18% were patients belonging to the SAVR group.

Regarding gender, 56% patients were men and 44% were women. No statistically significant differences with respect to gender have been detected.The mean age of the total sample was 80.6 [SD 4.1]. Statistically significant differences were detected in age, being higher in TAVR patients.

Statistically significant differences were detected, with a higher percentage of patients in the TAVR group with previous diagnosed of AF, Percutaneous Transluminal Coronary Angioplasty, Atrial Fibrillation, Anticoagulation Treatment, COPD, Anxiety-Depression Syndrome, low level of Hemoglobina, Glomerular Filtration and inferior Left Ventricular Ejection Fraction.

Statistically significant differences were detected, with a higher percentage of patients in the SAVR group who smoked.

**Table 1 Sociodemographic and clinical characteristics of the total sample.**

| | Total | Total HUMV | Total HUSA | Total TAVR | Total SAVR | Total SAVR | |
|---|---|---|---|---|---|---|---|
| | n = 227 | n = 186 | n = 41 | n = 113 | n = 114 | HUMV n = 73 | HUSA n = 41 |
| Gender | | | | | | | |
| Male | 128 (56%) | 100 (54%) | 28 (68%) | 59 (52%) | 69 (61%) | 41 (56%) | 28 (68%) |
| Female | 99 (44%) | 86 (46%) | 13 (32%) | 54 (48%) | 45 (39%) | 32 (44%) | 13 (32%) |
| | | | | p value* | | | |
| | | | | 0.21 | | | |
| | 80.6 | 80.8 | 79.6 | 82.8 | 78.4 | 77.6 | 79.6 |
| Age (years) | (SD 4.1) | (SD 4.4) | (SD 2.8) | (SD 4.1) | (SD 2.9) | (SD 2.5) | (SD 2.8) |
| | Median = 80.2 | Median = 80.4 | Median = 79.4 | Median = 83.4 | Median = 77.9 | Median = 77.4 | Median = 79.4 |
| | | | | p value** | | | |
| | | | | <0.01 | | | |
| | | | | | | p value* | |
| Civil status | | | | | | | |
| Married | 129 (57%) | | | 53 (47%) | 76 (67%) | | |
| Single | 15 (7%) | | | 7 (6%) | 8 (7%) | 0.01 | |
| Widowed | 80 (35%) | | | 51 (45%) | 29 (25%) | | |
| Separated/divorced | 3 (1%) | | | 2 (2%) | 1 (1%) | 0.01 | |
| Who do you live with? | | | | | | | |
| Alone | 42 (19%) | | | | | 23 (20%) | 0.04 |
| Family | 179 (79%) | | | 88 (78%) | 91 (80%) | | |
| Social-health care institution | 6 (2%) | | | 6 (5%) | – | | |
| Studies[1] | | | | | | | |
| Illiterate | 1 (0.4%) | | | – | 1 (1%) | | |
| No education | 64 (28%) | | | 39 (34%) | 25 (22%) | 0.19 | |
| Level 1 | 124 (55%) | | | 55 (49%) | 69 (60%) | | |
| Level 2 | 20 (9%) | | | 11 (10%) | 9 (8%) | | |
| <Level 3 | 18 (8%) | | | 8 (7%) | 10 (9%) | | |
| Cardiovascular risk factors | | | | | | | |
| HBP | 174 (77%) | | | 82 (72) | 92 (81%) | 0.15 | |
| DM | 73 (32%) | | | 34 (30) | 39 (34%) | 0.51 | |
| DLP | 149 (66%) | | | 76 (67%) | 73 (64%) | 0.61 | |
| Smoking | 68 (30%) | | | 31 (27%) | 37 (3%) | 0.02 | |
| BMI | 28.73 (SD = 4) | | | 28.70 (SD = 4.5) | 28.76 (SD = 3.5) | 0.91 | |
| Prior cardiac pathology | | | | | | | |
| Prior MI | 28 (12%) | | | 13 (11%) | 15 (13%) | 0.70 | |
| Prior PTCA | 49 (22%) | | | 35 (31%) | 14 (12%) | <0.01 | |
| Prior CABG | 2 (1%) | | | 2 (2%) | – | – | |
| Prior SAVR | 8 (3%) | | | 4 (3%) | 4 (3%) | 1 | |
| Prior MVR | 1 (0.4%) | | | – | 1 (1%) | – | |
| Other prior cardiac surgery | 6 (3%) | | | 3 (2%) | 3 (2%) | 1 | |
| Prior pacemaker | 14 (6%) | | | 7 (6%) | 7 (6%) | 0.99 | |
| Prior ICD | – | | | – | – | – | |
| Prior AF | 72 (32%) | | | 49 (43%) | 23 (20%) | <0.01 | |
| Previous non-cardiological pathology | | | | | | | |

(Continued)

| | Total | Total HUMV | Total HUSA | Total TAVR | Total SAVR | Total SAVR | |
|---|---|---|---|---|---|---|---|
| | $n = 227$ | $n = 186$ | $n = 41$ | $n = 113$ | $n = 114$ | HUMV $n = 73$ | HUSA $n = 41$ |
| Peripheral arteriopathy | 20 (9%) | | | 8 (7%) | 12 (10%) | 0.36 | |
| Prior stroke | 5 (2%) | | | 4 (3%) | 1 (1%) | 0.21 | |
| TIA | 6 (3%) | | | 3 (3%) | 3 (3%) | 1 | |
| Prior anticoagulation | 70 (31%) | | | 50 (44%) | 20 (17%) | <0.01 | |
| Prior COPD | 22 (10%) | | | 16 (14%) | 6 (5%) | 0.02 | |
| Anxiety-depression syndrome | 19 (8%) | | | 15 (13%) | 4 (3%) | <0.01 | |
| Cognitive impairment | 2 (1%) | | | 2 (2%) | 0 | – | |
| Prior liver disease | 11 (5%) | | | 6 (5%) | 5 (4%) | 0.74 | |
| Baseline situation | | | | | | $p$ value | |
| Dyspnea | 142 (63%) | | | 74 (65%) | 68 (60%) | 0.75[*] | |
| Angina | 15 (7%) | | | 8 (7%) | 7 (6 %) | – | |
| Syncope | 14 (6%) | | | 7 (6%) | 7 (6%) | – | |
| Dyspnea + Angina | 55 (24%) | | | 24 (21%) | 31 (27%) | – | |
| NYHA | | | | – | – | | |
| Class I | 19 (8%) | | | 4 (4%) | 15 (13 %) | | |
| Class II | 106 (47%) | | | 50 (44 %) | 56 (49 %) | <0.01[*] | |
| Class III | 79 (35%) | | | 40 (35%) | 39 (34 %) | | |
| Class IV | 23 (10%) | | | 19 (17%) | 4 (4%) | | |
| Prior Hb | 13 (DE 2) | | | 12.1 (DE 1.7) | 12.9 (DE 1.5) | <0.01[***] | |
| Prior GF | 66 (54–79) | | | 60 (48–75) | 70.1 (60–82) | <0.01[**] | |
| Prior LVEF | 55 (50–60) | | | 55 (50–60) | 58 (50–60) | 0.02[**] | |

**Notes:**

HUMV, Hospital Universitario Marqués de Valdecilla; HUSA, Hospital Universitario de Salamanca; SD, standard deviation; HBP, high blood pressure; DM, diabetes mellitus; DLP, dyslipemia; BMI, body mass index; MI, acute myocardial infarction; PTCA, percutaneous transluminal coronary angioplasty; CABG, coronary artery bypass grafting; SAVR, surgical aortic valve replacement; TAVR, transcatheter aortic valve replacement; MVR, mitral valve replacement; ICD, implantable cardioverter-defibrillator; AF, atrial fibrillation; TIA, transient ischemic attack; COPD, chronic obstructive pulmonary disease; NYHA, new york heart association; Hb, hemoglobin; GF, glomerular filtration; LVEF, left ventricular ejection fraction.

[1] Level 1: primary education or 5 years of EGB or equivalent; 2nd grade: Elementary Baccalaureate, School Graduate, complete EGB, Level 1 and 2 Vocational Training, Higher Baccalaureate; Level 3: Higher Technical School Degrees, University School Graduates and University College or University College Graduates, Bachelor's Degrees.

[*] Pearson's Chi-Squared Test.

[**] Wilcoxon test.

[***] Student's T-Test.

Statistically significant differences ($p < 0.01$) were detected in the NYHA scale between TAVR and SAVR, with a higher proportion of patients with NYHA in Class I in SAVR patients, and a higher proportion of NYHA in Class IV in TAVR patients.

There were a total of four (4%) patients in class I in the TAVR group *vs.* 15 (13%) patients in the SAVR group. There were 50 (44%) patients in class II compared to half of the patients in the SAVR group, 56 (49%). Also, 40 (35%) patients in the TAVR group were in class III *vs.* 39 (34%) patients in the SAVR group. Finally, 19 (17%) patients in the TAVR group were in functional class IV compared to 4 (4%) patients in the SAVR group.

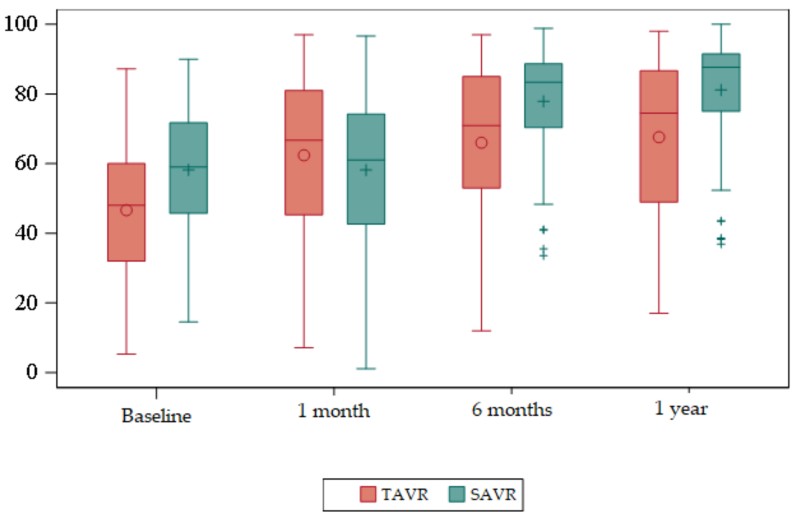

**Figure 2 Box plots about evolution of quality of life according to the SF-36 domains.** Red = TAVR Green = SAVR.

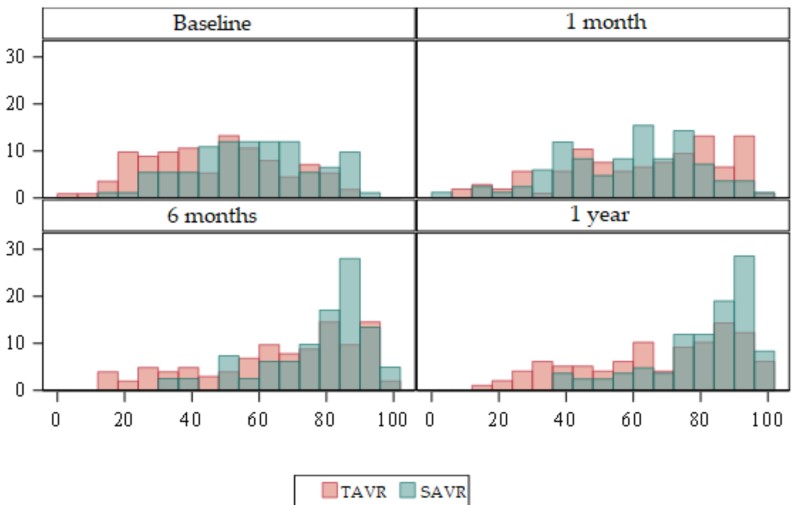

**Figure 3 Bar plots about evolution of quality of life according to the SF-36 domains.** Red = TAVR Green = SAVR.

## Evolution of quality of life

The 36-Item Short Form Health Survey questionnaire (SF-36) (Table S1) (Figs. 2 and 3).

Regarding the baseline quality of life score, the mean overall SF-36 in the TAVR group was 47, whereas in the SAVR group this was 58. Statistically significant differences were found ($p < 0.01$), with the mean overall SF-36 being higher in SAVR patients.

At 6 months and at 1 year statistically significant differences were observed ($p < 0.01$), with the mean being higher in the SAVR group compared to a mean in the TAVR group.

There were no statistically significant differences at 1 month.

According to the dimensions we obtained statistically significant differences in several areas.

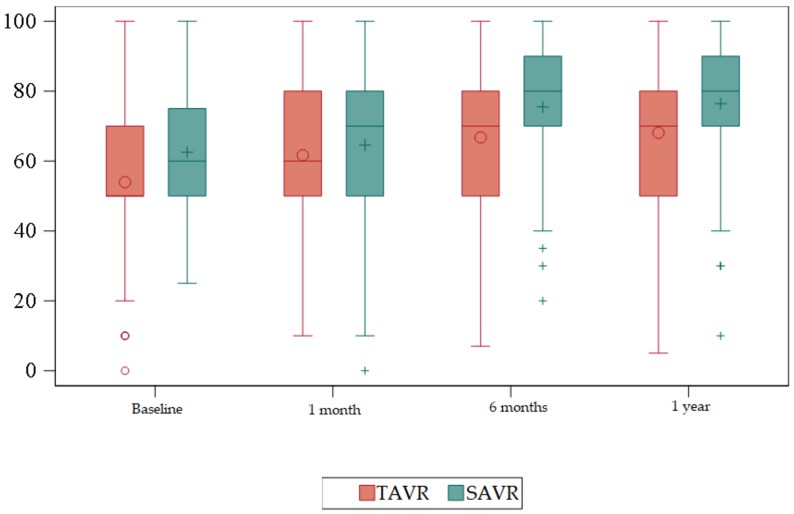

**Figure 4 Box plots about evolution of quality of life according to the EuroQol-5D.** Red = TAVR Green = SAVR.

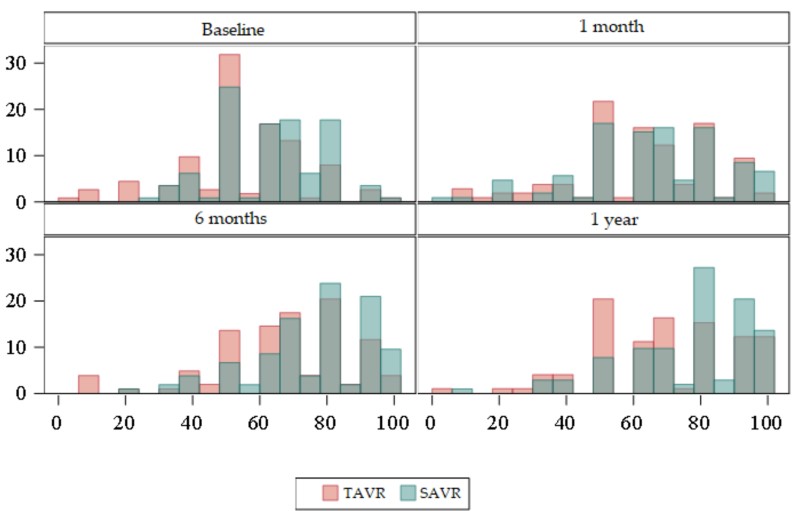

**Figure 5 Bar plots about evolution of quality of life according to the EuroQol-5D.** Red = TAVR Green = SAVR.

EuroQol five-dimension scale (EuroQol-5D) (Table S2) (Figs. 4 and 5)

Regarding the baseline quality of life score, the mean global EuroQol-5D in the TAVR group was 54 compared to 62 in SAVR. Statistically significant differences were detected ($p < 0.01$) with the mean global EuroQol-5D being higher in SAVR patients.

At 6 months, and at 1 year, statistically significant differences were detected ($p < 0.01$) with a higher mean in the SAVR group compared to a mean in the TAVR group.

There were no statistically significant differences at 1 month.

In relation to the evolution of quality of life by dimension, we obtained statistically significant results in Mobility at baseline ($p < 0.01$), at 6 months ($p < 0.01$) and at 1 year ($p < 0.01$) being higher in the SAVR group. At 6 months, 73 (70.9%) patients in the TAVR

**Table 2 Evolution of functionality at 1-year follow-up after the intervention.**

|  | Baseline | | 1 month | | 6 months | | 1 year | |
|---|---|---|---|---|---|---|---|---|
|  | TAVR n = 113 | SAVR n = 113 | TAVR n = 106 | SAVR n = 106 | TAVR n = 103 | SAVR n = 105 | TAVR n = 98 | SAVR n = 103 |
| Barthel scale | 91.5 | 97.1 | 92.7 | 92.5 | 91.7 | 97.4 | 92.2 | 98.2 |
|  | (IQR 90–100) | (IQR 95–100) | (IQR 90–100) | (IQR 90–100) | (IQR 90–100) | (IQR 100–100) | (IQR 90–100) | (IQR 100–100) |
|  | (SD 12.6) | (SD 6.3) | (SD 11.8) | (SD 15.3) | (SD 13.5) | (SD 6.9) | (SD 13.7) | (SD 5) |
|  | Median = 95 | Median = 100 | Median = 100 | Median = 100 | Median = 100 | Median = 100 | Median = 100 | Median = 100 |
| p value* | <0.01 | | 0.97 | | <0.01 | | <0.01 | |

Notes:
IQR, Interquartile Test; SD, Standar Deviation.
* Wilcoxon test.

group had no mobility problems compared to 97 (92%) in the SAVR group. In both groups this decreased slightly at 1 year, although the significant difference was maintained, with 71 (72.5%) patients in the TAVR group having no mobility problems compared to 93 (90%) in the SAVR group. There were no significant differences between the two groups at 1 month after the intervention.

Regarding Self Care, statistically significant differences were detected at baseline ($p < 0.01$), 6 months ($p < 0.01$) and 1 year ($p < 0.01$) being higher in the SAVR group.

As for Usual Activities, there were statistically significant differences in the four time points ($p < 0.01$) being higher in the SAVR group at baseline, 6 months and 1 year.

Regarding the Pain/Discomfort dimension, there were statistically significant differences at 1 month, with a $p < 0.01$ being higher in the SAVR group.

Finally, the Anxiety/Depression dimension showed statistically significant differences at baseline ($p < 0.01$), at 6 months ($p < 0.01$) and at 1 year ($p = 0.04$) being higher in the SAVR group.

### Evolution of functionality

Barthel Scale (Table 2) (Figs. 6 and 7).

Statistically significant differences were detected at baseline ($p < 0.01$), at 6 months ($p < 0.01$), and at 1 year ($p < 0.01$) being higher in the SAVR group.

The TAVR group showed an improvement in functionality, from a median of 95 prior to the procedure to a median of 100 in the remaining measurements during the year after the intervention.

## DISCUSSION

In our sample, statistically significant differences were observed regarding quality of life, measured using the SF-36 and EuroQol-5D, which was higher in patients who underwent SARV.

This difference was maintained in the quality of life at baseline, at 6 months and at 1 year. No statistically significant differences were found at 1 month after the procedure with any of the questionnaires, although the scores were lower in patients who underwent SAVR and this is justified by the characteristics of the procedure.

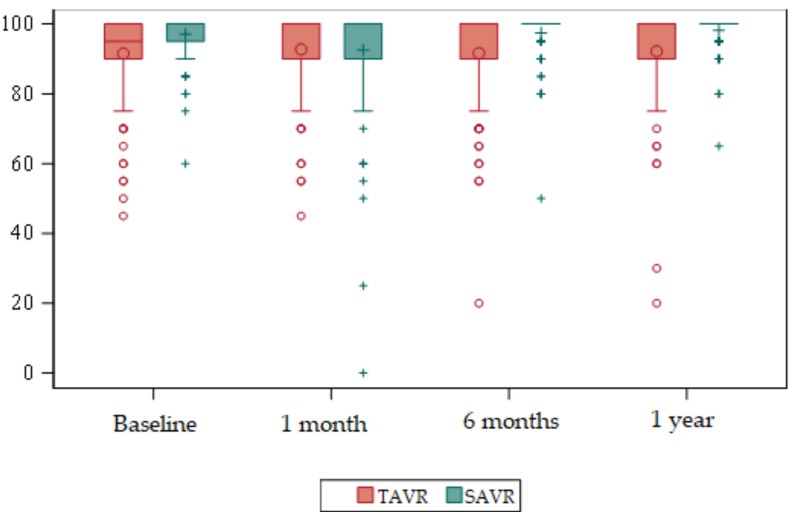

**Figure 6 Box plots about evolution of quality of life according to the Barthel Scale.** Red = TAVR
Green = SAVR.

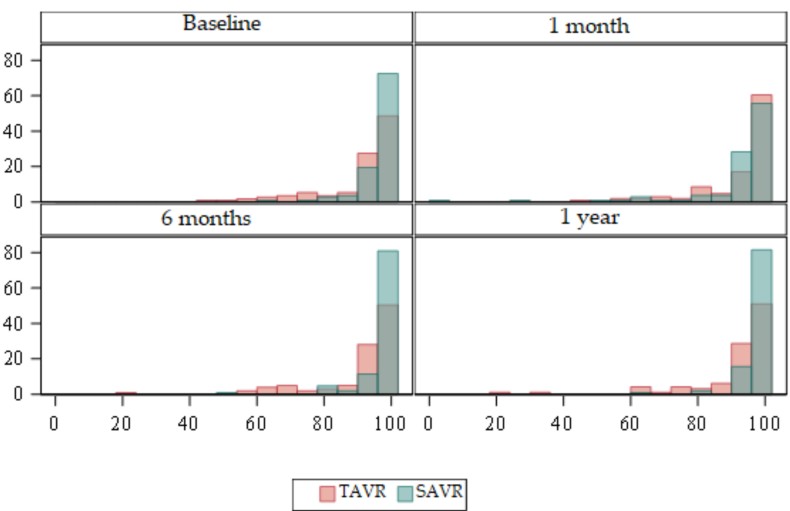

**Figure 7 Bar plots about evolution of quality of life according to the Barthel Scale.** Red = TAVR
Green = SAVR.

A better quality of life in patients undergoing TAVR can also be explained by the fact that the baseline quality of life (before the procedure) was higher and statistically significant in this group of patients. The baseline situation of patients who underwent TAVR was significantly worse since statistically significant differences were found in terms of anxious-depressive syndrome, previous AF, previous anticoagulation, and COPD, among others. Some studies suggest that COPD may represent an important factor long-terms in patients with aortic stenosis in relation to their mortality and quality of life (*Dziewierz et al., 2018*).

Statistically significant differences were also detected in The New York Heart Association Classification, with Class I being more frequent in patients undergoing SAVR

and Class IV in TAVR patients. All this may explain a higher baseline quality of life in patients undergoing SAVR.

These data differ from those obtained in the PARTNER trial (Reynolds et al., 2012) where the quality of life was superior in patients undergoing TAVR. This difference is mainly due to the selection process. In the PARTNER trial, the treatment applied was randomized (SARV vs. TAVR), whereas in our study, the usual practice was maintained, and the most appropriate treatment was decided in a clinical session after diagnostic catheterization. This meant that less aggressive approaches (TAVR) were decided in patients with a worse baseline situation and a higher surgical risk.

Our study enables us to analyze the evolution of the patients' quality of life by means of two questionnaires and by evaluating each item, obtaining highly relevant information. In the SF36 we detected differences in overall quality of life; however, when analyzing each item, these differences are not always maintained. At 1 month after the intervention, no statistically significant differences were found in "Physical Function" and "Bodily Pain", even though the SAVR procedure is much more aggressive. In the EuroQol-5D we observed that at 1 year, SAVR patients had better scores in all items except pain, where no differences were found.

Furthermore, it should be noted that the mean age of PARTNER patients (Reynolds et al., 2012) undergoing SAVR (84.8 (±6.5); 83.4(±5.5)) was appreciably higher than that of our sample (78.35 (±2.87)). However, although age is a factor to be considered, Thourani et al. (2015) concluded that nonagenarian patients undergoing TAVR in PARTNER 1 obtained an improvement in quality of life considerably superior to baseline as measured by the Kansas City Cardiomyopathy Questionnaire (KCCQ).

Although the overall score was higher in SAVR patients, both groups increased the overall SF36 by around 20 points. These results are confirmed by the EuroQol-5D where both groups increased their scores by 14 points at 1 year of follow-up. We cannot compare these results with PARTNER (Reynolds et al., 2012) since the results obtained in the quality of life questionnaire have not been published.

Our results are similar to those reported by other authors (Adams, Popma & Reardon, 2014; Durko et al., 2018; Gavalaki et al., 2020; Lauck et al., 2020; Surman et al., 2022). Adams, Popma & Reardon (2014); evaluated quality of life with KCCQ, obtaining slightly higher scores in patients with TAVR. Durko et al. (2018), in their sample started from very low quality of life scores (TAVR 36.6 (±9.8); SAVR 36.8 (±9.7)) and observed that the scores were similar at 1 year, with lower scores being obtained in those patients who had suffered a neurological complication. Tokarek et al. (2016), compared the quality of life in 163 patients who underwent TAVR and different aortic valve replacement procedures. Although its sample size was smaller, its results were similar to ours, obtaining a clear improvement in TAVR patients 1 month after the procedure and 1 year. Furthermore, they found no differences at 24 months.

Although variable findings related to quality of life are reported in the available studies, the literature shows greater cost-effectiveness, reduction of complications and longer life expectancy in patients who undergo TAVR (Adams, Popma & Reardon, 2014; Durko et al., 2018; Kerrigan et al., 2018; Reynolds et al., 2012; Thourani et al., 2015).

Regarding functionality, the patients who underwent TAVR began with lower scores with a median of 95 at baseline that reached 100 at 1 month with a mean of 92 that was maintained at 6 months and 1 year of follow-up. This means that even though the patients in our sample started at a worse baseline situation than the patients who underwent SAVR, they managed to improve their functionality after 1 month of the intervention and to maintain it, at least, until 1 year of follow-up. The EuroQol-5D item "Activities of Daily Living" showed statistically significant differences at 1 month after the procedure, with 51.89% of TAVR patients reporting that they had no problems performing these activities, compared to 33.96% of SAVR patients. However, at 1 year, up to 81.55% of SAVR patients reported having no problems performing daily activities.

This functionality scale is not usually included in studies that compare both techniques; however, studies that delve into cardiac rehabilitation in this population of patients report functionality data, before and after rehabilitation, with worse scores than ours. This shows that we can conclude that throughout the data collection, in general, our patients were very autonomous (*Ribeiro et al., 2017*).

*Adams, Popma & Reardon (2014)* studied the functionality of patients based on the number of activities of daily living affected. The study concluded that only 1.8% of patients undergoing TAVR had difficulties performing three activities of daily living compared to 5% of patients undergoing SAVR. *Olszewska-Turek et al. (2020)* compared quality of life, instrumental activities of daily living, and cognitive assessment in patients with TAVR and found no differences at baseline or at 13 months.

## LIMITATIONS

Our study has enabled us to analyze the daily practice of two hospitals of the Spanish Health Service with patients with aortic stenosis undergoing TAVR *vs.* SAVR; however, it has some limitations.

Firstly, the non-randomization of the patients in the treatment allocation has led to a bias, obtaining a sample of patients in the TAVR group with a significantly worse baseline situation.

The study was limited to two hospitals of the National Health Service, with 1 year of follow-up and patients over 74 years of age. The sample size was 227 patients and could be not sufficient to detect differences between groups.

In the Hospital Universitario de Salamanca (HUSA), only patients with SAVR procedure were recruited because the TAVR procedure was different, and therefore not comparable, with patients recruited in the Hospital Universitario Marqués de Valdecilla (HUMV).

Other limitation was that we use the generic QoL assessment tools intead cardiomyopathy specific questionnaire.

Follow-up was limited to 1 year, in future studies it would be relevant to extend follow-up and larger sample sizes to study phenomena such as the influence of COPD (*Dziewierz et al., 2018*) and the obesity paradox (*Tokarek et al., 2019*), among others.

Before the end of the 1-year follow-up, 15 patients in the TAVR group and 11 patients in the SAVR group died, representing a loss rate of 11.45%.

## CONCLUSIONS

The individualized choice of surgical approach, TAVR or SAVR, in patients with aortic stenosis improves patients' quality of life and function. Moreover, the TAVR procedure in patients with a worse baseline situation and a high surgical risk, manages to increase the quality of life and functionality of the patients by the same average points as the SAVR approach.

### Funding

This work was supported by the Fundación Instituto de Investigación Marqués de Valdecilla (IDIVAL) (Next-Val Program). The funders had no role in study design, data collection and analysis, decision to publish, or preparation of the manuscript.

### Grant Disclosures

The following grant information was disclosed by the authors:
Fundación Instituto de Investigación Marqués de Valdecilla (IDIVAL) (Next-Val Program).

### Competing Interests

The authors declare that they have no competing interests.

### Author Contributions

- Víctor Fradejas-Sastre conceived and designed the experiments, performed the experiments, analyzed the data, prepared figures and/or tables, authored or reviewed drafts of the article, and approved the final draft.
- Paula Parás-Bravo conceived and designed the experiments, analyzed the data, prepared figures and/or tables, authored or reviewed drafts of the article, and approved the final draft.
- Manuel Herrero-Montes performed the experiments, analyzed the data, authored or reviewed drafts of the article, and approved the final draft.
- María Paz-Zulueta performed the experiments, authored or reviewed drafts of the article, and approved the final draft.
- Ester Boixadera-Planas analyzed the data, authored or reviewed drafts of the article, and approved the final draft.
- Luis Manuel Fernández-Cacho conceived and designed the experiments, performed the experiments, prepared figures and/or tables, authored or reviewed drafts of the article, and approved the final draft.
- Gabriela Veiga-Fernández performed the experiments, analyzed the data, authored or reviewed drafts of the article, and approved the final draft.
- Maria Elena Arnáiz-García performed the experiments, authored or reviewed drafts of the article, and approved the final draft.

- Jose María De-la-Torre-Hernández conceived and designed the experiments, prepared figures and/or tables, authored or reviewed drafts of the article, and approved the final draft.

## Human Ethics

The following information was supplied relating to ethical approvals (*i.e.*, approving body and any reference numbers):

This study was approved by the Ethics and Clinical Research Committee on july, 11, 2014. (Protocol code: 22/2014)

## Data Availability

The raw measurements are available in the Supplemental Files.

## Supplemental Information

Supplemental information for this article can be found online at http://dx.doi.org/10.7717/peerj.16102#supplemental-information.

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
