# Peer review of "Surgical vs. transcatheter aortic valve replacement in patients over 75 years with aortic stenosis: sociodemographic profile, clinical characteristics, quality of life and functionality"

_PeerJ, doi:10.7717/peerj.16102_

## Round 0.1 · original submission · Major Revisions

Please address the following reviewer comments for making revisions to the manuscript.

Reviewer 1 ·

Basic reporting

The authors reported the comparison of SAVR and TAVR procedures in terms of quality of life (QoL) improvement after 1 year in a prospective study from 2 hospitals in Spain. The results provide new information for the current debate on SAVR vs TAVR. The authors already acknowledged some key limitations of this study. Language is clear and professional. However, there are some questions from the reviewer.

Experimental design

The authors claimed that "In HUSA, only patients with SAVR procedure were recruited because the TAVR procedure was different..." This is actually a critical point, which led to TAVR results solely from one hospital. Please explain the differences in detail.

Validity of the findings

Items (age, gender, etc) in Table 1 need to be statistically compared between total SAVR and TAVR and show the p value.
Given the mass amount of results, it is recommended that significantly different parameters in tables be highlighted/bolded.

Reviewer 2 ·

Basic reporting

This manuscript is well written and designed properly. Few questions:

- introduction is too long - need evaluation. Some more contemporary dat from current literature review might be helpful: https://pubmed.ncbi.nlm.nih.gov/34400917/
- sourc odf statistical software should be presented

Experimental design

-

Validity of the findings

-

Additional comments

- Despite low sample size this study another problem is utilization of generic QoL assessmnet tools - disease specific should be used such as MLHFQ or Kansas City Cardiomyopathy Questionnaire
Please discuss this in limitations section and add to discussion as comparsion with another studies with more comprehensive evaluation of QoL: https://pubmed.ncbi.nlm.nih.gov/26800644/
https://pubmed.ncbi.nlm.nih.gov/32368240/
- what new add this study to current knowledge? This topic is well descibed
-sample size is low, might be not sufficient to detect differences between groups
- since similar BMI between gropus, have aurthors tried to evaluate so called obesity paradox? please comapre your outcome in light of this study
https://pubmed.ncbi.nlm.nih.gov/30575008/
- higher rate of COPD was observed, some studies reported lower QoL in COPD patients, it seems to be important factor affecting long-term outcomes of patients with severe AS undergoing TAVI. - pleqase discuss this topic and potentital infuece on your outcomes: https://pubmed.ncbi.nlm.nih.gov/29185204/

Reviewer 3 ·

Basic reporting

no comment

Experimental design

no comment

Validity of the findings

no comment

Additional comments

The following manuscript is a retrospective study which investigates the pros and cons of receiving a SAVR or TAVR in patients over 75 years of age. The pool of patients are drawn from two Spanish hospitals. Broadly speaking, the study confirms much of what is already known regarding SAVR and TAVR: TAVR is generally recommended for older, high surgery risk patients while SAVR tends to be recommended for younger patients. Additionally, the overall outcomes are not surprising: TAVR may result in the same level of increases in patient quality of life compared to SAVR when the baseline of the patient condition pre-operation is taken to account. However, even despite the confirmation of previously known phenomenon, the study is still enlightening on local trends in Spain and may serve as additional datasets for future studies to be compared to. Overall, the paper is well written and easy to follow. However, there are several issues that the authors should address:

Major issues:
• There is a lot of information in tables 2 and 3. I think this information might be better represented as box and whisker plots or bar plots with standard deviation shown to be visually interpretable. I highly recommend the authors consider using more graphical representations of data throughout the manuscript as opposed to tabulating data. The tabulation of data is also useful in case readers are interested in the exact numbers. I recommend replacing the tables in the main body of the manuscript with plots and moving the tables to the supplemental information section.
• Much of the data is longitudinal data. I would highly recommend plotting longitudinal data as a line plot with the time on the x-axis and the quantity of interest (mean +/- STD) on the y-axis. Please do this for all the information in the tables. As is, the tables contain a lot of information that is tedious to look through.
• In the results section, the authors state numerous times that statistical significance was found between groups. However, I think it would be more informative to go a step further and state which group was higher/lower. Do this for all instances.
• There are many grammatical and spelling errors (too many to list out) within the manuscript. Would highly recommend authors do extensive proofreading.

Minor issues:
• Should cu-rative be curative?
• Line 69: willfocus should be will focus
• The wording of the sentence beginning in line 73 is awkward. Consider replacing it with the following: “Calcified or degenerative AS is the most frequent valve disease in the western world and requires the most intervention. It affects 25% of the population >65 years of age with 2-9% of this population experiencing some degree of AS. The prevalence of calcified for degenerative AS is expected to increase in the coming years with the population ageing.”.
• Line 89: should per-formed be performed?
• Line 89: the acronym STS is not defined. Please define it.
• In lines 102-107, I think the author’s motivation can be better explained. For instance, I would recommend including a statement about the age ranges studied in the PARTNER and COREVALVE studies to bring light to the fact that younger patients were likely given surgical valves whereas older patients were given transcatheter valves. Then, discuss about how older, high surgical risk patients are more likely to receive transcatheter valves. However, due to the markedly lower quality of life scores for older, high risk patients, their perception of an increase in quality of life post operation may be inflated. Therefore in the current study, we focus on studying patients within the same age range who received surgical vs transcatheter replacements to remove affects arising from age.
• Line 174: I would recommend changing the pronoun her to their to be gender neutral.
• Line 177-180: the sentences here sound very informal. They should be reworded to come across more professional and technically sound.
• Line 249: pa-tients should be patients

---

## Round 0.2 · accepted · Accept

The reviewers have acknowledged that all their comments have been addressed to satisfaction. I recommend that a final version be proofread to correct minor typographical errors. Congratulations!

Reviewer 1 ·

Basic reporting

The revision addressed all my comments. The manuscript is ready to be accepted.

Experimental design

The revision addressed all my comments. The manuscript is ready to be accepted.

Validity of the findings

The revision addressed all my comments. The manuscript is ready to be accepted.

Additional comments

The revision addressed all my comments. The manuscript is ready to be accepted.

Reviewer 2 ·

Basic reporting

In limitations section : Other limitation was that we use the generic QoL assessment tools "intead"cardiomyopathy specific questionnaire - should be "instead"?

No further comments. Authors replied in satisfactory manner. Despite limitations manuscript seems to be merit for publication

Experimental design

-

Validity of the findings

-

Additional comments

-

Reviewer 3 ·

Basic reporting

no comment

Experimental design

no comment

Validity of the findings

no comment

Additional comments

no comment